# Notions of Completeness in the EPR Discussion

**DOI:** 10.3390/e25040585

**Published:** 2023-03-29

**Authors:** Gerd Christian Krizek, Lukas Mairhofer

**Affiliations:** Department Applied Mathematics and Physics, University of Applied Sciences Technikum Wien, 1200 Vienna, Austria

**Keywords:** quantum mechanics, interpretation, EPR, Einstein–Podolsky–Rosen, entanglement

## Abstract

We explore the different notions of completeness applied in the EPR discussion following and amending the thorough analysis of Arthur Fine. To this aim, we propose a classification scheme for scientific theories that provides a methodology for analyzing the different levels at which interpretive approaches come into play. This allows us to contrast several concepts of completeness that operate on specific levels of the theory. We introduce the notion of *theory completeness* and compare it with the established notions of *Born completeness*, *Schrödinger completeness* and *bijective completeness*. We relate these notions to the recent concept of ψ-*completeness* and *predictable completeness*. The paper shows that the EPR argument contains conflicting versions of completeness. The confusion of these notions led to misunderstandings in the EPR debate and hindered its progress. Their clarification will thus contribute to recent debates on interpretational issues of quantum mechanics. Finally, we discuss the connection between the EPR paper and the Einstein–Rosen paper with regard to the question of completeness.

## 1. Introduction

Quantum mechanics is undoubtedly one of the most successful theories in physics, but its foundations and interpretations are still heavily disputed topics. The formalism of quantum mechanics allows for various interpretations, but it is surprising how strong the differences are with regard to the ontological and epistemological commitments of these interpretations [1,2].

Interpretational debates as part of the scientific method are as old as science itself. However, no question of interpretation in physics has attracted more attention than the discussion about the interpretations of quantum mechanics. The reason is certainly that it is more than simply an argument about different models about a physical situation. It is a controversy with pervasive philosophical implications and has one of its origins in the EPR debate on the completeness of quantum mechanics [3].

## 2. A Classification Scheme

To analyze the notions of completeness in the interpretations of quantum mechanics, we introduce a classification scheme that categorizes the structure of a physical theory—in particular, an interpretation of quantum mechanics.

The structure of scientific theories has been analyzed in the literature in several accounts from positivism to structuralism [4,5,6,7,8,9,10,11].

The presented classification scheme does not claim to add new conceptual aspects on the nature of theories in the philosophy of science; the levels could be defined differently and the boundaries between the levels could be chosen differently, but the proposed scheme is valuable for contrasting the content of different interpretations of quantum mechanics.

A physical theory consists roughly of the mathematical formalism and an interpretation, which contains the definition of symbols, theoretical terms, measurement assignments, concepts and principles, and in some cases an ontology. We refine this coarse breakdown into a four-level classification, where the notion of levels does not entail a strict hierarchy.

**Definition** **1.**
*Level 1 of a physical theory consists of a set of mathematical quantities. These mathematical quantities are related to each other by mathematical formalism.*


**Definition** **2.**
*Level 2 of a physical theory relates a set of measurement assignments to the mathematical quantities of Level 1. It describes how the defined quantities in the mathematical description of the theory are related to sensations and experience. It relates mathematical objects to theoretical terms and experimental practice.*


**Definition** **3.**
*Level 3 of a physical theory contains a number of concepts and principles that are related to the mathematical description of the theory. These principles can be formulated as statements or logical expressions.*


**Definition** **4.**
*Level 4 of a physical theory involves the ontology of the theory and the metaphysical entities or concepts that are introduced by the theory.*


We make use of the following figurative elements in Figure 1 to visualize the classification scheme.

By dots, we depict a physical quantity, which can be the property of a physical entity or any other physical measure that is introduced by the theory. On Level 1 of the classification scheme, a physical quantity is basically a mathematical symbol. These dots are related by mathematical relations, the laws between the physical quantities. On the first level, these are the mathematical laws that are fulfilled by the mathematical symbols representing the physical quantities.

On the second level, the physical quantities receive their names and relation to empirical results, the measurement assignments. By this, the pure mathematical formalism becomes a physical law. The physical quantities and the laws involve concepts that are set up by theoretical terms and governed by principles. These are depicted by the areas spanned by some physical quantities and laws, and belong to Level 3 of the classification scheme.

On the ontological level, Level 4, there are entities that are a priori unknown. A physical theory may provide impicit or explicit statements on the ontology, statements that provide an insight into the metaphysical reality behind the empirical content. These statements involve usually entities that are claimed to exist independently of our perception; we denote them as elements of objective reality and depict them by a diamond.

Applying these figurative elements to the classification scheme, we can give an overview of the scheme in Table 1.

We will apply the proposed scheme to analyze the notions of completeness in the EPR argument.

## 3. Completeness in the EPR Debate and Beyond

The completeness used in the EPR argument and accompanying sources is a peculiar concept. At first glance, it seems to be clear what the authors of the EPR paper meant by completeness; however, we will demonstrate that there are several notions of completeness involved. The discussion of their role and relevance in the context of the EPR argument requires a clear definition of different notions of completeness.

### 3.1. Theory Completeness

In 1934, Einstein explained his understanding of the term completeness [12]:


*“A complete system of theoretical physics consists of concepts and basic laws to interrelate those concepts and of consequences to be derived by logical deduction.”*


In Section 2, we presented a notion of the structure of a physical theory, by using physical quantities, the laws connecting them, and concepts to add structure and principles to the theory. Here, we understand these elements in the Machian sense so that they are not connected, per se, to an ontology. Einstein emphasized that these elements are “… free inventions of the human mind…” [12] and therefore pure constructions of thought. Therefore, this definition of completeness rather is the demand for consistency of the theoretical system. In any sense, it forgoes to provide ontological statements. We will address this completeness by the term *theory completeness*. Figure 2 shows a theory that is complete in this sense.

An application of *theory completeness* is given in [13]:


*“One of the imperfections of the original relativistic theory of gravitation was that as a field theory it was not complete; it introduced the independent postulate that the law of motion of a particle is given by the equation of the geodesic. A complete field theory knows only fields and not the concepts of particle and motion. For these must not exist independently of the field but are to be treated as part of it.”*


We wish to provide a definition for *theory completeness* that reflects as closely as possible the concept that Einstein had in mind.

**Definition** **5.**
*A theory fulfills theory completeness if its physical quantities and laws are connected by concepts that relate only to each other, and the theory renounces extra assumptions of concepts that exceed the framework of the theory.*


*Theory completeness* does not refer to an ontology. It only refers to elements of the theory, and demands the consistency of the used concepts. The concept of closedness (Vollständigkeit) used in [13] can be seen as identical to *theory completeness*. As can be seen in Figure 3 Einstein clarified in the handwritten manuscript of the ER paper which concept of completeness he had in mind in the specific paragraph.

References to *theory completeness* can be found in [12] (pp. 164, 166); [15] (p. 778); [13] (pp. 76, 77); [5] (pp. 316, 344); [16] (pp. 320, 323).

### 3.2. Bijective Completeness

The EPR paper questions the completeness of quantum mechanics, but is it *theory completeness* that is challenged? If we refer to the EPR paper, a definition of completeness is given [15]:


*“Whatever the meaning assigned to the term complete, the following requirement for a complete theory seems to be a necessary one: every element of the physical reality must have a counterpart in the physical theory”*


As can be seen immediately in the graphical representation of the classification scheme in Figure 4, this definition of completeness differs from the definition of *theory completeness*; depending on the definition of physical reality, it may even be read to extend the term completeness to the ontological level, whereas the *theory completeness* definition clearly refers only to the theory and its elements. The completeness criterion of the EPR paper relates the elements of reality to the elements of a theory.

From an accompanying source, one further aspect of the notion of completeness in the EPR paper can be seen. Karl Popper received a letter from Einstein in September of 1935, where Einstein explained the idea of the EPR paper in a different way [17] (p. 413):


*“Since a complete description of a physical state must necessarily be an unambiguous description (apart from superficialities such as units, choice of the co-ordinates etc.), it is therefore not possible to regard the ψ-function as the complete description of the state of the system.”*


Einstein concludes that a complete description has to be unambiguous, so, if every element of the physical reality has to have a unique correspondence in the theory, it must be vice versa. This unambiguousness assumption is not stated explicitly in the formulations of criteria of reality, completeness or separability/locality. However, it implicitly plays a crucial role in the EPR argument [18]:

*“Man möchte nun gerne folgendes sagen:* Ψ *ist dem wirklichen Zustand des wirklichen Systems eindeutig zugeordnet. […] Wenn dies geht rede ich von einer vollständigen Beschreibung der Wirklichkeit durch die Theorie.”*

*“One would now very much like to say the following:* Ψ *stands in a one-to-one correspondence with the real state o f the real system. […] If this works, I talk about a complete description of reality by the theory.”* (Translation by Arthur Fine [19] (p. 71))

Ref. [19] calls this completeness *bijective completeness*. Following [15], a definition would be as follows.

**Definition** **6.**
*Bijective completeness means that every element of the physical reality $\epsilon_{i}$ must have a counterpart $t_{i}$ in the physical theory.*


Though *bijective completeness* is a bijection between elements of the theory and elements of reality as can be seen in Figure 5, it starts with the latter ones. Starting with elements of reality causes the problem that the elements of reality have to be identified a priori in their relation to the corresponding elements of the theory. For this purpose, [15] had to assume a criterion of reality to identify these elements of reality.

It is further interesting that Einstein seems to have seen a problem with an argument based purely on *bijective completeness* in the EPR paper. Superficialities such as the choice of coordinates would not fulfill *bijective completeness*, as Einstein clarifies in his letter to Popper [17] (p. 413).

Einstein gives no exact definition of completeness in the EPR paper, but states that Definition 6 of *bijective completeness* is a necessary requirement for his concept of completeness. Is it possible that *theory completeness* is the completeness that Einstein had in mind? *Theory completeness* is only defined by elements of the theory, and makes no statement about the elements of reality that the theory refers to; therefore, it can be ruled out that *bijective completeness* is a necessary requirement of *theory completeness*. Thus, *theory completeness* is not the completeness that Einstein had in mind, since he introduced *bijective completeness* as a necessary requirement. While *bijective completeness* is applied in the EPR argument, it is not the only sense in which Einstein challenged the completeness of quantum mechanics.

References to *bijective completeness* can be found in [15] (pp. 777, 778, 780) and [5] (p. 341).

### 3.3. Born Completeness

Since *theory completeness* cannot be identified with the completeness addressed in the EPR paper, there must be a different definition of completeness. Ref. [19] identifies two further concepts of completeness in the EPR debate: *Born completeness* and *Schrödinger completeness*.


*“According to the Bornian concept, a complete description is essentially nonprobabilistic; genuinely probabilistic assertions are necessarily incomplete.”*


These concepts of completeness can be identified in letters from Einstein to Schrödinger, where Einstein also mentioned the “Einstein’s Boxes” Gedankenexperiment, which discusses the probabilities of finding a ball in a box [20] (letter 206, p. 537). Both concepts go beyond the scope of *theory completeness* and concern the relation between the theory and the predicted phenomena. Ref. [21] introduces the notion of *predictable completeness* to describe this relation. A theory might predict all phenomena observed in experiments correctly without yet grasping their fine structure. Such a theory would not be considered predictable complete and a re-evaluation of the experimental data would lead to the discovery of additional variables governing this fine structure. Both *Born completeness* and *Schrödinger completeness* imply *predictable completeness*.

**Definition** **7.**
*Born completeness is fulfilled for a theory if the theory contains no probabilistic elements. A theory that contains statistical accounts and probabilistic statements is considered to be Born incomplete.*


Following this definition, quantum mechanics is incomplete per definition. This requires no further arguments.

Is *Born completeness* the completeness that Einstein meant in the EPR paper? If *Born completeness* was addressed, it would imply that the authors of the EPR paper had in mind that without *bijective completeness*, *Born completeness* could not be true. This follows from the EPR condition of completeness.

On the other hand, the statement that “quantum mechanics fulfills *Born completeness*” is false. In this case, from rules for the material conditional, it follows that independent of the truth value of the *bijective completeness*, the statement that “*theory completeness* is a necessary requirement for *Born completeness*” is true. To show that the truth value of *bijective completeness* is therefore meaningless, we can rule out that *Born completeness* is meant as the conception of completeness in the EPR paper.

References to *Born completeness* can be found in [5] (pp. 339, 341, 343, 347).

### 3.4. Schr ödinger Completeness

Ref. [19] (p. 71) gives an account of *Schrödinger completeness*:


*“By contrast the Schrödinger view is that probabilities can be fundamental, not to be reduced to something else. Thus the Schrödinger conception is that a complete description of a state of affairs can be a probabilistic assertion, with probability less than unity, which (somehow) tells the whole truth about that state of affairs. If there were some further truth to be told, then the probabilistic assertion would be an incomplete description.”*


The definitions of probability in context with *Born completeness* and *Schrödinger completeness* differ vastly. Whereas the concept of the probability of occurrence of outcomes in an ensemble of events is used in the first conception, the latter defines probability as fundamental and related to single events. The interpretation of probability is beyond the scope of this work. We will therefore apply an operational definition of *Schrödinger completeness* that is based on Einstein’s considerations in his letter to Schrödinger [20] (letter 206, p. 537).

**Definition** **8.**
*A theory containing probabilistic statements about elements of the theory is Schrödinger complete if those statements refer to single events or entities and are fundamental in the sense that nothing more can be said about the system.*

*The theory is considered to be incomplete if there are potential new elements in the theory that would abandon the probabilistic character of the theory.*


Einstein calls these factors “fremde Faktoren” (external factors). The translation of “fremd” is peculiar, since, in the literal translation, it would mean “strange”, “weird”, “alien” or “different.” However, in context, it refers to factors that are not considered in the theory. They are not part of the theory, so we decide to use the non-literal translation “external”. The definition of *Schrödinger completeness* refers to what we would nowadays call "hidden variables."

Carefully analyzing the argument in the letter to Schrödinger from the 19th of June 1935, we find another reference to Einstein’s notion of completeness [20] (letter 206, p. 537). Here, Einstein rephrases Bohr’s point of view as follows:


*“Nun beschreibe ich einen Zustand so: Die Wahrscheinlichkeit dafür, daß die Kugel in der ersten Schachtel ist, ist $\frac{1}{2}$. Ist dies eine vollständige Beschreibung?"*


*“I describe a state this way: The probability for a ball being in the first box is $\frac{1}{2}$. Is that a complete description?"* (Translation by the author)

From this description of a single event, Einstein switches to his own ensemble interpretation:


*“Der Zustand vor dem Aufklappen ist durch die Zahl $\frac{1}{2}$ vollständig charakterisiert, deren Sinn sich bei Vornahme von Beobachtungen allerdings nur als statistischer Befund manifestiert."*


*“The state before opening the box is completely characterised by the number $\frac{1}{2}$, whose meaning becomes manifest as statistical account by applying measurements on the system."* (Translation by the author)

If we interpret this complete characterization of the state by the number $\frac{1}{2}$ in the sense of *bijective completeness*, the number is ascribed to an element of reality. Einstein interprets this number in an epistemic way as probability. Depending on the interpretation of probability, these elements of reality would represent propensity, knowledge about the state of the ball or degrees of belief.

In principle, this position entails a description that would fulfill *bijective completeness*, because one can give a bijection to elements of reality in each of these accounts. Therefore, it is plausible to identify *Schrödinger completeness* as the completeness that Einstein challenged mainly in the EPR paper, even though he used several other notions of completeness in the EPR argument.

### 3.5. ψ-Completeness

In an account to classify hidden variable models, Ref. [22] (p. 131) proposed a definition of *ψ-completeness*:

*“An ontological model is ψ-complete if the ontic state space* Λ *is isomorphic to the projective Hilbert space PH (the space of rays of Hilbert space) and if every preparation procedure $P_{\psi }$ associated in quantum theory with a given ray ψ is associated in the ontological model with a Dirac delta function centered at the ontic state $\Lambda_{\Psi }$ that is isomorphic to* Ψ*, $p\left(\lambda |P_{\psi }\right)=\delta \left(\lambda -\lambda_{\psi }\right)$.”*

Ref. [22] (p. 147) claims that *ψ-completeness* is identical to *bijective completeness*:


*“It is quite clear that by “real state of the real system”, Einstein is referring to the ontic state pertaining to a system. Bearing this in mind, his definition of completeness can be identified as precisely our notion of ψ-completeness given in Definition 2”*


However, we believe that the classification scheme proposed above might help in distinguishing the meaning of completeness on the different levels of a theory. *ψ-completeness* identifies the metaphysical entities of Level 4 as the mathematical entities of Level 1. This is a misunderstanding of Einstein’s view on completeness and elements of reality. While such an identification is possible, Einstein did not adhere this mathematical realism.

Einstein does not use the phrase “elements of reality” in his letter to Schrödinger [20] (letter 206, p. 538), but refers to *bijective completeness*:


*“Man beschreibt in der Quantentheorie einen wirklichen Zustand eines Systems durch eine normierte Funktion ψ der Koordinaten (des Konfigurationsraumes). Die zeitliche Änderung ist durch die Schrödinger-Gleichung eindeutig gegeben. Man möchte nun gerne folgendes sagen: ψ ist dem wirklichen Zustand des wirklichen Systems eindeutig zugeordnet. Der statistische Charakter der Meßergebnisse fällt ausschließlich auf das Konto der Messapparate bzw. des Prozesses der Messung. Wenn dies geht rede ich von einer vollständigen Beschreibung der Wirklichkeit durch die Theorie. Wenn aber eine solche Interpretation nicht durchführbar ist, nenne ich die theoretische Beschreibung unvollständig.”*


*“In quantum theory a real state of a system is described by a normed function ψ of coordinates (of configuration space). The evolution in time is given by Schrödinger equation uniquely. One would like to say: ψ is corresponding to the real state of the real system uniquely. The statistical character of the measurement results is a consequence of the measurement apparatus respectively the measurement process. If this works, I speak about a complete description of reality by a theory. If such an interpretation is not possible, I call the theoretical description incomplete.”* (Translation by the author)

Ref. [22] concludes from these statements that Einstein’s *bijective completeness* is identical to their *ψ-completeness*, but the latter one sets up an isomorphism between the projective Hilbert space and an ontic state space, whereas *bijective completeness* is a bijection between elements of a theory and elements of reality. These elements of reality have no defined mathematical structure, per se. In our proposed classification scheme for interpretations of quantum mechanics, the elements of reality are located in Level 4, the ontological level. Assuming that the elements of reality are elements of an ontic state space is a kind of mathematical realism, but we believe that this is not the intention behind the claim of Einstein.

On the other hand, Ref. [22] has good reasons to assume that Einstein, by referring to the “real state of the real system” [20] (letter 206, p. 538), refers to elements of a theory that would replace quantum mechanics. They propose an argument for Einstein’s preference for the EPR argument, in contrast to his incompleteness argument from 1927. According to them, it was Einstein’s goal to emphasize the epistemic character of the state vector and to look out for hidden variable theories.

*ψ-completeness* is a useful classification tool for hidden variable theories, but we refuse to identify *ψ-completeness* and *bijective completeness* in general based on the involvement of different structures in these two conceptions. This does not affect any applications of *ψ-completeness* in [22].

### 3.6. Remarks on Completeness and the EPR-ER Connection

The targets of the EPR paper are manifold; one aim is to show that *Schrödinger completeness* does not hold for the unitary dynamics of quantum systems. Einstein claimed that the statistical character of quantum mechanics did not result from the measurement process. The unitary evolution of the state vector itself is, according to him, incomplete in the sense of *Schrödinger completeness*. The criticism of quantum mechanics is a story, but more interesting in our context is the motivation for asserting this incompleteness. Replacing a successful theory with a new unknown theory without a good reason encounters opposition. Einstein had to convince the community of the incompleteness of quantum mechanics to present a new program [15]:


*“While we have thus shown that the wave function does not provide a complete description of the physical reality, we left open the question of whether or not such a description exists. We believe, however, that such a theory is possible.”*


Einstein retreated quickly, already in 1936, to the position of *Born completeness*, which to him was sufficient to show the need for a theory that fulfills his requirements. Referring to his article [16], he notes in a letter to Besso [23] (p. 403),


*“Es freut mich dass Du meinen kleinen Aufsatz gelesen hast. Hast Du auch gemerkt wie unlogisch Pauli darauf geantwortet hat? Er leugnet es, dass diese Art der Beschreibung unvollständig sei, sagt aber im selben Atemzuge, dass die ψ Funktion eine statistische Beschreibung des Systems sei, die Beschreibung einer System-Gesamtheit. Dies ist doch nur eine andere Form der Aussage. Die Beschreibung des (individuellen) Einzelsystems ist unvollständig!”*


*“I am pleased to hear that you read my little article. Have you recognised also how illogical Pauli responded? He denies that this description is incomplete, but states in the same breath that the ψ function is a statistical description of the system, the description of a system-totality. This is just a different formulation of the statement: The description of an (individual) single-system is incomplete!”* (Translation by the author)

This statement is the essence of *Born completeness*. It shows that the EPR arguments and all its successor arguments by Einstein had one main purpose: to show the incompleteness of quantum mechanics for the specific purpose of justifying the need for a new field theory to replace quantum mechanics. Despite the ambiguous notions of completeness, it seems fair to conclude that it was the idea to present this new complete theory in the next published work following two months after the EPR paper; the Einstein–Rosen bridge paper.

We have presented the meaning of the terms “completeness” and “reality” and pointed out which aspects of reality were especially relevant to Einstein in the context of the 1935 papers. In the following section, we will apply these clarifications to discuss the connection of the 1935 papers in the context of the unified field theory program.

### 3.7. Completeness and the Unified Field Theory Program

Einstein had in mind to resolve the incomplete statistical character of quantum theory with a unified field theory (the completeness refers to *Born completeness*). Even in 1934, one of the conclusive statements of his paper on the method of theoretical physics addresses the statistical character [12]:


*“I still believe in the possibility of giving a model of reality, a theory, that is to say, which shall represent events themselves and not merely the probability of their occurrence.”*


We have seen above that in Einstein’s opinion, the concept of completeness is strongly opposed to the statistical character of quantum theory. In the original EPR argument in [15], he used the reality criterion, locality (Trennungsprinzip—separation principle) and the contradiction principle to conclude the incompleteness of quantum mechanics. In [5], Einstein no longer refers to these principles. Instead, he argues for the ensemble interpretation, which is an indirect application of *Born completeness*. According to the ensemble interpretation, quantum mechanics only refers to a statistical ensemble of events, not to a single event itself. To Einstein, this argument was sufficient to show the incomplete character of quantum mechanics and to head for an underlying theory. We can see that this is one of the main agendas of the EPR paper [15]:


*“While we have thus shown that the wave function does not provide a complete description of the physical reality, we left open the question of whether or not such a description exists. We believe, however, that such a theory is possible.”*


This new theory, which he presented in the ER-paper, at first independent of the EPR critique, was meant to solve the problem of the atomistic character of matter and fulfills Einstein’s requirement of completeness [13]:

*“In favor of the theory (The Einstein-Rosen bridge model for elementary particles) one can say that it explains the atomistic character of matter as well as the circumstance that there exist no negative neutral masses, that it introduces no new variables other than the $g_{\mu \nu }$ and $\phi_{\mu \nu }$, and that in principle it can claim to be complete (or closed). On the other hand one does not see a priori whether the theory contains the quantum phenomena.”*(our underlining)

Here, he means *theory completeness*, which is evident when he places the term “closed” in brackets to underline that it refers to the closedness of a unified field theory that hopefully solves the particle problem.

It is remarkable that Einstein distinguishes between the atomistic character of the theory and the question of whether the theory contains quantum phenomena. The “atomistic character” of matter requires a sharp localization of the physical entities, while, at the same time, Einstein wants to avoid singularities of the field. The Einstein–Rosen bridge would have solved this in a way satisfactory to Einstein. It is not exactly clear what Einstein understands by the term “quantum phenomena” and how the theory could contain them. It is comprehensible that he means the quantum states and the formalism of quantum mechanics, which cannot be found in the Einstein–Rosen bridge representation for a particle, but it certainly was Einstein’s hope that the quantum effects would emerge out of this new model.

In [5], he presented the EPR argument in a modified form in line with the ER bridge. Here, the connection of these two papers as part of a program is evident:


*“Es wird aber gezeigt, dass die Überzeugung von der Unfähigkeit der Feldtheorie, diese Probleme mit ihren Methoden zu 1ösen, auf Vorurteilen beruht.”*



*“It will be shown that the conviction of the inability of field theory to solve these problems (The problems refer to the inability of field theory to give account for a molecular structure and to describe quantum phenomena with methods of field theory [5] (p. 347)) rests on a prejudice.”*


In addition,


*“Angesichts dieser Sachlage erscheint es mir durchaus gerechtfertigt, die Frage ernsthaft zu erwägen, ob nicht doch die Grundlage der Feldphysik mit den Quanten-Tatsachen vereinbar ist.”*


*“In view of this situation it seems appropriate to me to reconsider the question if a unification of the foundations of field theory and quantum facts is possible.”* (Translation by the author)

This sums up the scientific program that Einstein attempted to present. At the beginning, he refers to a comparison between quantum mechanics and statistical mechanics, and states that a statistical or effective theory is not an appropriate starting point for developing a complete (in the sense of *Born completeness* and *theory completeness*) theory. In his view, quantum mechanics is this effective theory that delivers only statistical predictions, but gives no account of the single events.

In a more private environment, a letter from the 16th February 1936 to his friend Michele Besso, Einstein expresses this point of view explicitly. Since, to our knowledge, this letter has not been cited in the context of the EPR discussion, we wish to cite it in full (we omit a passage on the political situation in Europe) [23] (p. 308).


*“Ich halte die statistische Physik trotz all ihrer Erfolge doch für eine vorübergehende Phase und habe Hoffnung, zu einer wirklich befriedigenden Theorie der Materie zu gelangen. Ich sende Dir gleichzeitig eine kurze Arbeit, die den ersten Schritt darstellt. Das neutrale und das elektrische Teilchen erscheinen gewissermassen als Loch im Raume, derart, dass das metrische Feld in sich selbst zurückkehrt. Der Raum wird als zweischalig dargestellt. In der Schwarzschild’schen strengen zentralsymmetrischen Lösung erscheint das Teilchen im gewöhnlichen Raume als Singularität vom Typus $1-\frac{2m}{r}$. Durch die Substitution $1-2m=u^{2}$ wird das Feld regulär im r-Raume. Wandert u von −∞ bis +∞, so wandert r von +∞ zu r=2m und hierauf wieder zurück zu r=+∞. So kommen beide "Blätter" im Riemann’schen Sinne zustande, die an der “Brücke” r=2m bezw. u=0 stetig zusammenhängen. Aehnlich bei der Elektrizität. Die Aufgabe an der ich mit einem jungen Kollegen (russischer Jude) unablässig schwitze ist die Behandlung des Mehrkörperproblems auf dieser Basis. Wir haben aber die ernsthaften Schwierigkeiten des Problems bereits überwunden, sodass sich bald zeigen wird was daran ist. Jedenfalls ist es eine wundervolle mathematische Aufgabe.”*


*“To me statistical physics despite its success is a transitory phase and I have hope that we arrive at a really satisfying theory of matter. I send you enclosed a short work, that represents a first step. The neutral and the electrical particle appear as a hole in space, of this kind that the metric field returns to itself. Space is represented as two sheets. In the strict spherical symmetric solution of Schwarzschild the particle appears in usual space as singularity of the kind $1-\frac{2m}{r}$. By substitution $1-2m=u^{2}$ the field becomes regular in r-space. If u goes from −∞ to +∞, r is going from +∞ to r=2m and back again to r=+∞. This represents both “sheets”, that are connected by the “bridge” at r=2m respectively u=0. Likewise as it is in electricity. The challenge a young college (russian jew) and I labour away over is the many-body-problem on that basis. We have conquered the serious problems already, so it will show soon if there is something serious about it. Anyway it is a wonderful mathematical problem.”* Translation by the author

The connection between Einstein’s critique of quantum mechanics—specifically its statistical character as well as the problems of the localizability of particles—and the connection to the Einstein–Rosen bridge approach could not be closer than presented here.

Einstein’s unified field theory program failed for reasons we will not discuss in this work. For a comprehensible presentation of this field theoretic program, refer to [24]. Einstein expressed his hopes that the program would succeed and how it might contain the quantum phenomena [5]:


*“Erst die Untersuchung des Mehr-Brücken-Problems kann zeigen, ob diese theoretische Methode eine Erklärung für die empirisch erwiesene Massengleichheit der Teilchen in der Natur liefert, und ob sie den von der Quantenmechanik so wunderbar erfassten Tatsachen gerecht wird.”*


*“Only the examination of the many-bridge-problem can show if this theoretical model provides an explanation for the empirical proven equality of masses of particles in nature, and if it can reproduce the facts that are represented by Quantum Mechanics in such a delightful way.”* (Translation by the author)

These hopes of Einstein seem to be maintained in recent program claims that reconsider the idea of a deep connection between entanglement and Einstein–Rosen bridge solutions. Recent claims bring back the EPR-ER idea in the context of quantum gravity [25]:


*“General relativity contains solutions in which two distant black holes are connected through the interior via a wormhole, or Einstein-Rosen bridge. These solutions can be interpreted as maximally entangled states of two black holes that form a complex EPR pair. We suggest that similar bridges might be present for more general entangled states.”*


It is remarkable that the critique by [26] sounds not wholly different from Einstein’s reservations on quantum mechanics:


*“Now I feel that our current views of Quantum Mechanics are provisional; it’s the best we can do without a much deeper understanding of its connection with gravity, but it’s not final. The reason involves a very particular development, the so called ER=EPR principle. ER=EPR tells us that the immensely complicated network of entangled subsystems that comprises the universe is also an immensely complicated (and technically complex) network of Einstein-Rosen bridges.”*


The term “provisional” and the statement that quantum mechanics is not to be seen as final can be understood as synonymous to Einstein’s claims for the incompleteness of quantum mechanics, in the sense of *theory completeness*.

## 4. Conclusions

It is Einstein’s merit that he raised questions about the foundations of quantum mechanics that are still relevant and hinder contemporary physics. Against this background, it is more than a question of the history of science to rethink the old debates. The EPR paper contributes an argument that is still relevant to the discussions on the interpretation of quantum mechanics.

We have presented several notions of completeness that can be identified in the EPR paper, the ER paper and its accompanying sources: *theory completeness*, which refers only to the elements of the theory; *bijective completeness*, or the condition of completeness, as it was denoted in the EPR paper; *Born completeness*; and *Schrödinger completeness*. The recent conception of *ψ-completeness* amends the presentation. We argue that *ψ-completeness* is not identical to *bijective completeness* based on the types of elements that both notions of completeness refer to.

Einstein’s intention in writing the EPR paper was to demonstrate the incompleteness of quantum mechanics in the sense of *Schrödinger completeness*, to justify his ambitions to look for a complete theory in the sense of *theory completeness*. This new theory should fulfill *theory completeness* and *Born completeness*, as well as explain the atomistic character of microscopic entities. This turned out to require the singularity-free description of particles by fields. Furthermore, it has to reproduce the quantum phenomena, which are successfully described by the formalism of quantum mechanics.

If one browses through Einstein’s papers of the year 1935, the ER paper coincides with the EPR paper, and a closer look reveals that they are closely connected. With the ER paper, Einstein presented this new theory, which developed into his attempts to present a unified field theory. His critique on quantum mechanics was in parallel with his ongoing work on the unified field theory, but later focused on a position that was easier to hold. He criticized *Born completeness*, because to him it was undeniable that a new theory was needed, and he shifted his effort from criticizing quantum mechanics to developing a unified field theory.

The tempting thought to explain nonlocal phenomena inside quantum mechanics with ER bridges is nothing that Einstein would have considered, because he used separability to show that quantum mechanics is incomplete, in the sense of *Schrödinger completeness* independent of a measurement carried out on the system.

By applying the separation principle and excluding the measurement process from consideration, Einstein aimed to show that the statistical character of quantum mechanics is not owed to the measurement process, but resides only in the state vector formalism and the Schrödinger evolution. The state vector formalism is fully deterministic, and by this he concludes that the formalism of quantum mechanics is incomplete.

In the literature on the EPR argument, the statistical character of quantum mechanics plays a key role in Einstein’s critique of quantum mechanics, and it truly deserves this role. We emphasize one further essential motivation for Einstein’s attempt to reconstruct quantum mechanics, the “atomistic structure”, or, as we identify it, the problem of the absolute localization of particles. In classical field theory, attempts to obtain the localization of particles result in field singularities. For Einstein, this problem was an incompleteness in general relativity, in the sense of *theory completeness*. He aimed to solve this problem with a new complete theory, which he first presented in outlines in the ER paper. The ER bridge was intended merely to solve the problem of localization of particles, but it also would provide a singularity-free representation of particles and, according to Einstein, would be consistent with Heisenberg’s uncertainty principle. It must have been insights such as this that encouraged him to hope that this approach promised that quantum phenomena might emerge in his new theory and, by this, reconstruct quantum mechanics out of field theory. Unfortunately, these hopes of Einstein failed, and until the end of his life, he could not present a satisfying solution to this problem.

Recent programmatic approaches that claim the equivalence of EPR=ER principles have been presented. We cannot estimate how the EPR=ER approaches will contribute to the development of a unified theory of quantum gravity, but the idea is promising, and if the program turned out to be successful, it would offer a late gratification for Einstein’s field theory program to unify quantum mechanics and gravity, even if this success was not in the originally intended direction. It would also underline the extent of Einstein’s intuition, which brought forward foundational problems in the interpretation of quantum mechanics and probably indicated a promising path in this quest.

## Figures and Tables

**Figure 1 entropy-25-00585-f001:**
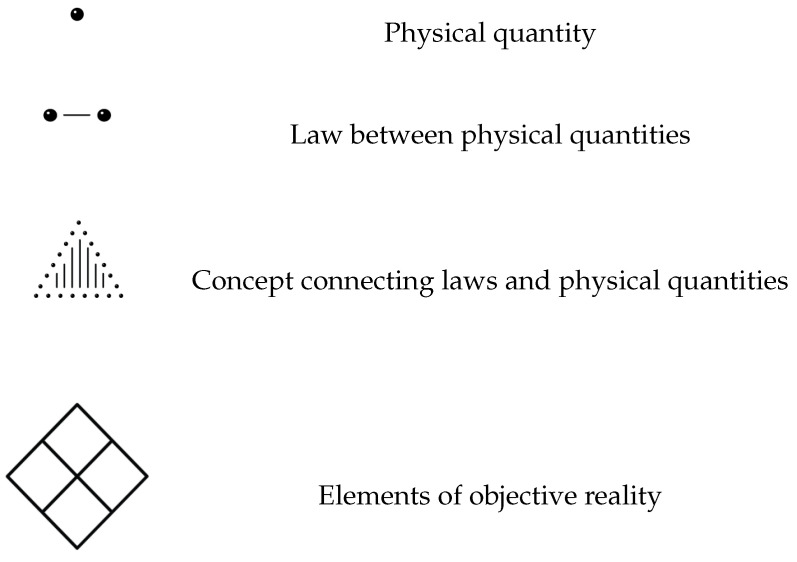
Figurative elements used in the classification scheme.

**Figure 2 entropy-25-00585-f002:**
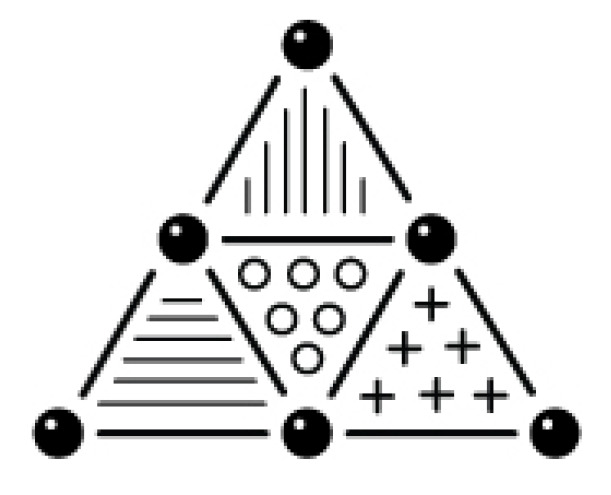
Depiction of a theory fulfilling *theory completeness*: Laws (bold lines) relate mathematical entities (bold dots), which obtain physical meaning from the principles of the theory (areas).

**Figure 3 entropy-25-00585-f003:**
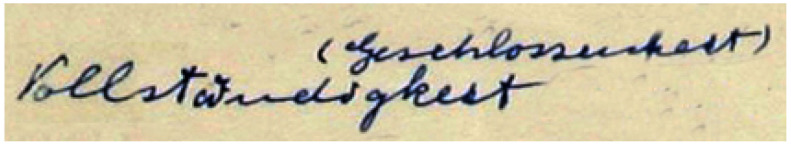
Completeness —(Closedness) [14].

**Figure 4 entropy-25-00585-f004:**
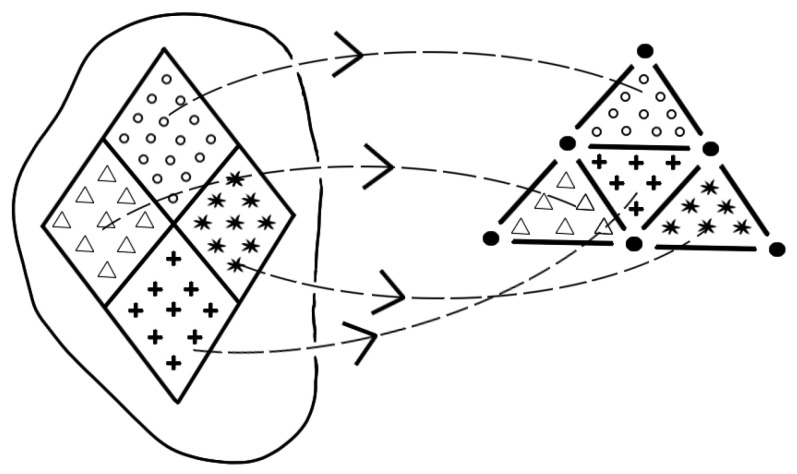
The completeness requirement of the EPR paper: An injective relation from the elements of reality (left side) to the elements of reality (right side).

**Figure 5 entropy-25-00585-f005:**
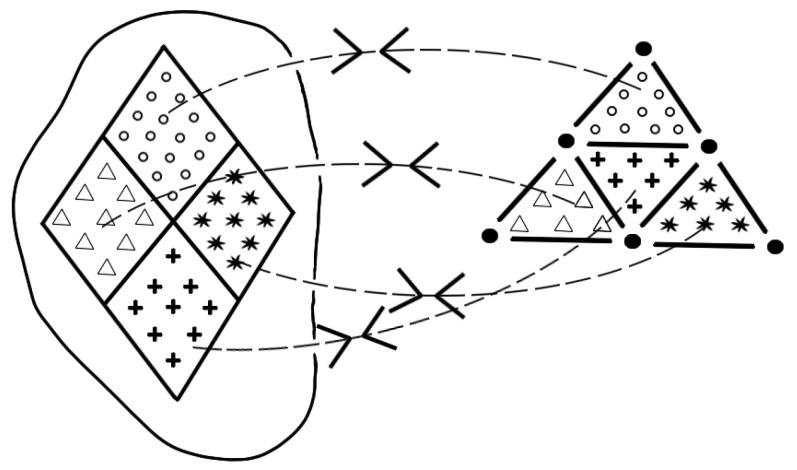
*Bijective completeness* resulting from an unambiguousness relation between the elements of reality (left side) and the elements of the theory (right side).

**Table 1 entropy-25-00585-t001:** Classification scheme: We distinguish four levels of a theory to provide a categorization to analyze different notions of completeness.

1	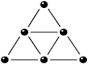	Mathematical lawsbetween mathematical symbols
2	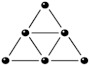	Interpretation of physical quantitiesRelation of physical quantities and measurementsMeasurement laws
3	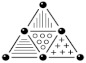	Concepts and principles
4	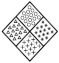	Ontology—beables—elements of reality

## Data Availability

Not applicable.

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
