# Peer review of "Notions of Completeness in the EPR Discussion"

_entropy, 2023, doi:10.3390/e25040585_

Round 1

Reviewer 1 Report

This paper carries out a detailed examination of the notion of completeness that is central to the argument of the EPR paper. The authors first argue that EPR blur the distinction between “theory completeness” and “bijective completeness”, which are not the same as each other. They then remark that the goal of EPR was to demonstrate the incompleteness of QM relative to the standard of Schrodinger completeness and suggest that the new theory they were advocating was one that would conform to both theory and Born completeness (but not bijective completeness). Finally they argue that the notion of psi-completeness recently introduced by Spekkens and his group is not equivalent to the notion of theory and/or bijective completeness, as maintained by Spekkens et al. (although this detracts in no way from their program).

Irrespective of whether one agrees with the various viewpoints advanced by the authors, I think they have performed a valuable service by putting together this list of alternative interpretations of completeness, backed up with relevant quotes by their proponents. I found myself learning a lot from the original sources and the later commentaries on them.

The later part of the paper is concerned with the EPR-ER connection and the current pursuit of this idea by physicists interested in quantum gravity. I was not aware that the EPR and ER papers, written in the same year, were partly motivated by the shared goal of finding a unified field theory as Einstein conceived it. The ideas advanced in refs.[17] and [18], which propose that black holes can exist in entangled states via ER bridges, represents a remarkable synthesis of the ideas in the EPR and ER papers of 1935. Of course, this line of work still remains to be developed more fully and it will be interesting to see where it leads.  

I do of course have my own reactions to some of the views advanced by the authors, but don’t think this is the place to discuss them. I think other readers are also likely to be stimulated by this article as I have been, and want to enter into a further discussion of these matters with the authors. That, in my opinion, is sufficient reason to recommend publication of this paper.

Author Response

We thank reviewer 1 for his thorough reading of the manuscript and for his positive review.

Reviewer 2 Report

This manuscript discusses notions of completeness in the EPR paper, with input from some other Einstein documents.  The abstract claims that the EPR argument contains conflicting notions of completeness, which presumably would be exposed in the text itself.  However, the conflicts discussed in the text do not affect the EPR argument in any way.

The EPR argument indeed involves ontic elements, "elements of physical reality," and therefore necessarily goes beyond the "theory completeness" of section 3.1.  This is often emphasized by using the term "realism" in discussions of EPR.

The notion of "Born completeness" (which strikes me as rather awkward terminology, given that it stands in direct contradiction to the Born rule) of course conflicts with "Schroedinger completeness," but the first is essentially another terminology for determinism, whereas the latter implies indeterminism.  As quoted in the text itself, just before Fig. 3, the EPR argument does not assume one or the other of these: it begins with "Whatever the meaning assigned to the term complete, ..." and continues logically and clearly.  It is indeed a pity, and a source of misunderstandings, that the notion of determinism (sometimes called complete causality) is confused with the completeness of a theory, or with the notion of realism.  But the present text does not clarify this issue.

Another apparent conflict is discussed in section 3.5, but again, it is only indirectly related to the EPR argument.  In fact, it relates to the "levels" of a theory defined in section 2, and specifically to the way that the diagrams introduced there seem to imply that the ontic level 4 is somehow more closely related to the concepts of level 3 (see, e.g., the hatching of the triangles and the diamonds in Fig. 3).  In contrast, in the EPR argument, an ontic element of physical reality would correspond to a statement such as "the momentum of particle A is 17 units," which would indeed seem to relate to level 1.  Perhaps the approach of Ref. [14], which attempts to develop mathematical terminology to discuss level 4 as well, is not misplaced after all.

I must also remark that section 2 is rather vague, gives no references at all, and does not clarify how the suggested system of levels relates to previous discussions of physical theories and their interpretations in the philosophy-of-science literature.  This is not appropriate for a publication in Entropy.

In the later sections of the article, the relation with the ER paper is discussed, but it is inappropriate to do so without discussing other aspects of the EPR argument.  In particular, regardless of the issues of completeness, an essential part of the EPR paper shows that a separable theory which reproduces the predictions of QM for entangled particles must be deterministic (see, e.g., Bell, 1981).  And this involves measurements -- making a prediction for the position of particle A involves first making a position measurement on particle B and then using the value obtained in that measurement to make the specific prediction (which could be verified by a second measurement, of particle A).  Unfortunately, the fact that such a theory cannot exist at all -- that separability (aka local causality) contradicts agreement with QM -- could not have been known to Einstein, as Bell's Theorem is a much later development.  (In this sense, Born completeness does appear as a conclusion of a part of the EPR argument, at least for completely entangled particles; however, this is not the completeness of the theory but a completeness of the predictions given by the theory, in the sense that the predicted values are fully determined, not statistical.)

This would cast a different light on some of the statements made later, e.g., "Einstein retreated" on line 309, or "excluding the measurement process" on line 493.  Furthermore, section 4 more generally provides a discussion of possible implications rather than a conclusion which could be drawn on the basis of argumentation given earlier.

I therefore recommend that the manuscript be rejected.  I hope that the authors will nevertheless find some of the above remarks illuminating.

Author Response

We thank reviewer 2 for his statements.

The reviewer points out, that the different notions of completeness are intimately related with the questions of causality and realism. This is precisely why we argue that the interpretation of the term completeness strongly affects the EPR argument, contrary to the reviewer's statement that this was in no way the case. Nevertheless, we think that these notions should not be confused and the question of what completeness of a theory means deserves an evaluation of its own.

The reviewer reads the differentiation of the levels on which a theory plays out as the introduction of a hierarchy of those levels. This is unfortunate, as we tried to state a clear as possible that such a hierarchy is not implied and discuss several possible relations between those levels. Thus, an approach such as Reference [14] for discussing the ontology of a theory in might indeed be fruitful, and the authors have emphasized this in the text. However, Einstein did not attempt such a solution.

The reviewer remarks, that section 2 of the paper lacks references and should relate the system of levels of a theory to previous discussions of physical theories and their interpretations in the philosophy-of-science literature.

The authors have developed the system themselves to the purpose of analyzing the notions of completeness in the EPR argument and further aspects of the interpretations of Quantum Mechanics. An extensive review of the literature of structuring theories has been provided in a pre-print version of the article but has been removed due to the length of the article. We have now included these references thanks to the remark of the reviewer.

The reviewer states that a discussion of the relation between the EPR and the ER paper would benefit from including the notions of causality and reality. This of course is true but exceeds the scope of the paper.

The reviewer differentiates between the predictions of a theory and the theory itself. While this is useful with many respects, we do not agree to the strict separation between these two aspects that the reviewer seems to imply. For example, the falsification of a physical theory occurs via the demonstration of a conflict between predicted measurement outcomes and obtained measurement outcomes. Indeed, it the predictions of a theory might well be understood as part of the ontological level of the theory. The reviewer writes him/herself: "...an ontic element of physical reality would correspond to a statement such as 'the momentum of particle A is 17 units.' " This can as well be read as a statement about a measurement outcome as about an ontic property of particle A. More generally speaking, that "the predicted values are fully determined, not statistical" (reviewer 2) seems to us to be as much a statement about the theory as about the predictions of the theory. Nevertheless, the reviewers' remarks indicate that we have not made clear enough the relation between a theory and its predictions with regard to the question of completeness. We have clarified this point and included a reference to the paper "Seventy Years of the EPR Paradox" by Marian Kupczynski which introduces the notion of predictable completeness.

Reviewer 3 Report

The article "Notions of Completeness in the EPR discussion" examines the notions of completeness that were most probably at hand in Einstein's mind around the time of publication of his celebrated 1935 manuscript. Through the citation of various works and letters by Einstein, Rosen, Podolsky, Popper, Schrödinger and others, the authors eloquently show that various notions of completeness with differing can be identified. In doing so, they demonstrate that the notion of completeness involved in these exchanges was not unique.

While certainly of interest, I do not see a link between the topic at hand here and entropies. Apart from a description of concepts, I also can't identify in the manuscript the result of a scientific research. Therefore, I believe that the topic of this article is not in line with the scope of the 'entropy' journal, see https://www.mdpi.com/journal/entropy/about. Other venues, including one of the 400+ journals of MDPI such as Histories of Philosophies would be more suitable.

Author Response

We thank reviewer 3 for his work. His/her main objection is that the paper does not fit the scope of entropy. However, we submitted this paper as contribution to a special issue on completeness. Furthermore, entropy has published a wide range of papers on issues of foundations of physics and the philosophy and history of science. We thus think that entropy is well suited for the publication of the paper.

Round 2

Reviewer 3 Report

I apologize to the authors for the mis-judgment in my first report as I didn't notice that the article was to appear in a special issue focusing on the completeness of quantum theory. This article, discussing the question of completeness specifically in the EPR argument, is obviously very suited to this issue, and will certainly constitute a valuable piece of it.